# Evaluation of Bonding Gap Control Methods for an Epoxy Adhesive Joint of Carbon Fiber Tubes and Aluminum Alloy Inserts

**DOI:** 10.3390/ma14081977

**Published:** 2021-04-15

**Authors:** Witold Rządkowski, Jan Tracz, Adam Cisowski, Kamil Gardyjas, Hubert Groen, Marek Palka, Michał Kowalik

**Affiliations:** Institute of Aeronautics and Applied Mechanics, Warsaw University of Technology, 00-665 Warsaw, Poland; adamcisowski@wutracing.pl (A.C.); kamilgardyjas@wutracing.pl (K.G.); hubertgroen@wutracing.pl (H.G.); mpalka@meil.pw.edu.pl (M.P.); michal.kowalik@pw.edu.pl (M.K.)

**Keywords:** adhesive joint, bond gap control, epoxy adhesive, glass beads, carbon fiber tubes, aluminum alloy inserts

## Abstract

The aim of this paper is to compare two methods of epoxy adhesive bond gap control: one with a geometrical (mechanical) solution and the other with glass beads, which have the diameter of the desired bond gap and are mixed with an epoxy adhesive. The adhered materials were carbon fiber composite tubes and aluminum alloy inserts, which were used as wishbones in a suspension system of a motorsport vehicle. It was assumed that the gap thickness would be equal to 0.2 mm and the length of a bond would be 30 mm. The internal diameter of the tubes was 14 mm and 18 mm, whereas the inserts’ external diameter was 13.6 mm and 17.6 mm. Their surface has been subjected to mechanical treatment with sand paper starting from 240 grit up to 400. The adhesives used were EA 3425 and EA 9466 cured at 80 °C for 2 h. The results showed that the glass beads method provides more consistent and better results as compared to the geometrical (mechanical) method. Further study in the area of fatigue and interfacial failure modes could be useful.

## 1. Introduction

This paper discusses and validates the design process of an adhesive connection between two engineering materials: carbon fiber composite tubes and aluminum alloy inserts, which create wishbones of a motorsport vehicle. The technological aspects of the connection will be addressed, especially the method of controlling the gap and different adhesive products used to bond the parts together. The design of the wishbones will not be discussed fully; however, one of the main design features is that the loads in the system occur only in the axis of the tubes, i.e., the tubes and inserts are only subjected to tension or compression along the axis. Such design can be obtained by careful geometrical design of the whole suspension system. There were many approaches to such designs made by other motorsports teams [1,2,3,4]; more general knowledge about such systems is available in [5,6].

The connection of reinforced polymers to metals can be realized in a various ways. One of them is a bolted connection or riveted connection type, which is applicable for some cases. However, as discussed in a review [7,8], there are some downsides to this solution such as delamination during drilling of the holes, which could lead to failure of heavily loaded components such as wishbones discussed in this paper. Therefore, an adhesive connection will be used, since the axial forces present in the design will result in shearing of the adhesive bond, which is the optimum working direction of adhesive. Usually, the adhesive connection between carbon and metal is performed using epoxy-based adhesives, which are cured to obtain full strength. This paper focuses on two epoxy adhesives: EA 9466 and EA 3425 and their applicability for final wishbone structure.

The design process consisted of determining the geometry of the adhesive bond basing both on available adhesive models as well as real life testing and the technological solutions applied. This means that the area that will be focused on in this research is practical adhesion. This involves many issues, which need to be addressed in an adhesive bond for it to fulfil its goal. The mentioned issues are, e.g., surface treatments, adhesive type, and the curing process. Some of these have been described in a guide [9,10] together with other aspects of adhesion. The key conclusion is that the adhesion is a property of a system; thus, results obtained from other geometries and designs might not be comparable. This is crucial since the strength values given by manufacturers are based on tests performed with given standards, which are accurate yet might not produce the same results on other geometries. Therefore, the design must be validated by real life testing, which should answer the question which epoxy adhesive is optimal for the final wishbone as well as determine the proper gap control method. Having said that, the results obtained from experiments will be contrasted with the achievements of other teams in wishbone design and testing [11,12].

The aim of this study is to determine which of two bond gap control methods will be more suitable for the designed bond used in the wishbone project of a motorsport vehicle. The two investigated methods of controlling the gap were the following:The geometrical method (also referred as mechanical) is based on two surfaces with a radius increased by 0.2 mm at the end and beginning of the insert adhesive surface, which results in a grove for the adhesive surface on the aluminum insert.Glass beads added to the adhesive, which have a diameter of 0.2 mm and fill the gap between adherends, consequently locking the radial movement.

There are also other methods of bond gap control [13] such as wire spacers and film adhesives, yet they cannot be applied in this design as inserts must be slid into the tubes. The method used in both [11,12] was a geometrical (mechanical) method which is based on additional surfaces to control the bond gap. In this research, a novel approach will be used with glass beads mixed with the adhesive. This approach might compromise the bond performance slightly, but overall it can be superior to a mechanical solution as it could result in a lower mass of a joint as the additional surfaces are no longer necessary.

There have been studies on the gap control methods and adhesives containing additives such as particles of nano-Al_2_O_3_ [14], but in this case the particles are larger and are not added to increase the strength of a bond. As far as a geometrical solution is considered, there has been research on geometrical patterns [15] which impact the fracture toughness of a joint, which is important, although there the focus is on the technological aspect rather than the pattern or surface preparation. The value of the bond gap thickness and its influence on strength was a topic of discussion in a few studies [16,17,18], and the main conclusion was that the increased thickness of a gap decreases the strength of a bond, which contradictory with the analytical models such as that of Goland-Reisner. In general, the topic is still controversial and further studies are necessary. Thus, in this study it was decided to set the bond thickness to 0.2 mm based on some recommendations from the technical support of the adhesive manufacturer as well as the results of similar research performed [11,12]. This is due to the fact that the scope of this research is not to determine the optimum bonding gap, but rather to focus on the technological aspect of bond control as well as to determine the adhesive which will perform the best out of the two selected.

## 2. Materials and Methods

### 2.1. Materials Used

Two adhered materials were carbon fiber tubes and aluminum alloy inserts. The tubes are commercially available woven composite rods acquired from [19]. Although they were bought, it is also possible to manufacture them by using preimpregnated fabrics of carbon fiber and glass fiber as seen in the tutorial in [20]. The carbon fiber used in the tubes is Torray 700 high modulus unidirectional fiber (Easy composites Ltd., Stoke-on-Trent, UK), and the glass fiber is unidirectional glass (Easy composites Ltd., Stoke-on-Trent, UK). It is important to notice that the tube layup consists of 5 layers of plies. The axial layers (carbon fiber) are made from unidirectional material as they carry the main loads. The radial layers (glass fiber) are oriented at about 90 degrees and improve the crush strength of the final product. All parameters of the tubes are stored in the Table 1.

The aluminum inserts are machined out of stock 7075 aluminum alloy rods (Oberon, Warsaw, Poland). This kind of aluminum should be commercially available in most of material suppliers. The thermal treatment type was T-651, but it should not influence any of the adhesion properties, only mechanical properties such as ultimate tensile strength, etc.

The epoxy adhesives used were two commercially available products: EA 9466 and EA 3425 (Henkel-Adhesives, Warsaw, Poland). They are typical 2-part adhesives (resin and hardener), which require mixing before application. Both EA 9466 and EA 3425 can be cured at room temperature, yet elevated temperature increases their final strength properties by approximately 10% if cured for 2 h at 80 °C [21,22]. Some of the parameters, important from the point of view of this study, were stored in Table 2. The full range of properties can be found in the technical data sheets of the products.

Lastly, the method of controlling the bond gap was realized with 0.2 mm commercially available glass beads acquired from [23]. The recommended mass ratio of beads to adhesive is about 2%. The beads were added to mixed adhesive and stirred in a small container that was later placed in a vacuum chamber to degas the mixture. However, for an industrial scale of application the method should be changed to a premixed solution of hardener and beads as it would not decrease the working time (pot life) of the adhesive.

### 2.2. Designing Bond Geometry

The forces acting on the structure were calculated from vehicle kinematics. The limiting condition in case of connecting rods was the buckling condition. With all the safety coefficients, the diameter of the upper wishbone tubes must be at least 15 mm with a wall thickness of 1 mm and for the lower wishbone where the forces are higher—20 mm × 1.2 mm. However, such diameters are not available commercially; thus, the closest to them were chosen, i.e., 16.7 mm × 1.35 mm and 20.8 mm × 1.4 mm. This results in an inner diameter (ID) of the tube to be equal to 14 mm and 18 mm, respectively. Consequently, it establishes the radial dimension of a designed bond, leaving the length and gap to be determined.

#### 2.2.1. Adhesive Bond Models

To determine the length and gap of a bond, adhesive models can be used as well as experimental results. As described in the Introduction, there are some discrepancies between them as real-life testing does not match the theoretical assumptions [13]. Nevertheless, it is good to know them for design purposes, especially their advantages and disadvantages which will be described later. The first of the models is the Kendall Model [24], which is possibly the most basic one as it does not even consider the adhesive. It allows one to determine the force needed to peel the joint apart. It assumes that only the surface energy is holding the joint together; thus, the results are underapproximated.

A more sophisticated bond model is needed to approximate the initial geometry of a bond. This can be carried out with the Goland-Reissner bond model. The model is more complicated in terms of mathematics, yet there are very good calculators created by [9,25], which can visualize the stresses in the joint. Looking at them, it can be noticed that there are a few parameters that influence the strength of the design:Adherend material height, stiffness, and Poisson ratio. Higher values of both parameters cause less peel stress at the end of a bond since stiffer material deflects less when loaded with a force or bending moment. Therefore, there is less off-plane loading of adhesive joints (i.e., peeling of a bond). Fortunately, the stiffness of a composite tube is around 90GPA, which is a relatively high value, yet its thickness is quite small—1.35 mm.Adhesive height (bond gap). Higher values of bond gap result in smaller values more smooth distribution of both shear and peel stresses. This would indicate that the bonding gap should be as large as possible. This, however, contradicts some of the experimental research.Adhesive stiffness (E) and Poisson ratio. Higher stiffness of the epoxy increases the values of stresses at the joint ends. Thus, it would be reasonable to use as compliant an epoxy as possible. On the other hand, such epoxy leads to higher deformations, which is not ideal in a suspension system of a vehicle.Finally, the length of the adhesion does not influence the strength as much as the width, since most of the stress is concentrated at the beginning and end of a joint. Thus, increasing this parameter adds a lot of mass into the final structure without a valuable increase in bond strength. Thus, it is better to design a wider joint instead.

#### 2.2.2. Adhesive Bond Model—Experimental Results

The last step of the theoretical design focused on researching the results obtained by other teams. There has been an interesting approach made by the Harbin Racing Team [11], where the results showed that the ultimate tensile force for a tested bond was at maximum when the bond gap was 0.2 mm. Additionally, the length of a bond increased its strength linearly. Similar results were obtained by the Massachusetts Institute of Technology Team [12], where the optimum bonding gap was about 0.2 mm (0.008 inch to be exact). Both studies used a geometrical method of bond control, where there was a surface which determined the bond gap thickness. For this research, we decided to use the findings of those two teams in terms of the gap thickness, yet we changed the method of controlling it to beads with a diameter equal to the desired bond gap thickness. Although the adhesive used was different, it can be assumed that it was an equivalent of EA9466 based on the performance stated by the manufacturer. Additionally, the control samples with a geometrical method of controlling the gap will be created to be able to compare the adhesives. The value of a 0.2 mm gap is also recommended by the glass bead manufacturer for epoxy-based adhesives. The length of a bond was set to 30 mm for all samples at it was reported to have the ultimate strength needed for the project and could be compared to other results. All models used can be summarized using their advantages and disadvantages as described in Table 3.

### 2.3. Bond Gap Control

Knowing all geometrical parameters of the designed bond, the test samples can be prepared. The specimens for the tests were manufactured using conventional machining methods such as turning and milling. The idea of both the geometrical and bead solutions of bond gap thickness control has been depicted in Figure 1 and Figure 2. The tolerances for diameter were between 0 and −0.02 mm, and the tolerance for a 30 mm-long adhesive surface was ±0.1 mm for all samples

Although only the bigger inserts were shown (for tubes with an internal diameter of 18 mm), the smaller ones (14 mm) have similar dimensions. The only difference is that the diameter of the adhesive surface is equal to 13.6 mm, and in case of the geometrical solution the surfaces for the bond gap control thickness have diameter of 14 mm. When preparing the samples, the adhesive has been applied in great excess to both the inside of the tube and the surface of the inserts, and then the inserts have been slid into tubes. During the cure cycle, they have been fixed with a set of vice clamps. The specimens were prepared in a set of 8 for each type of geometry/adhesive.

### 2.4. Surface Preparation and Curing

The preparation of the surface is a very broad topic in adhesion and bonding. There are various techniques that can be used [26,27] to prepare the surface. One should remember that the surface preparation can influence the shear strength and fatigue life of a bond [28,29]. Since all combinations cannot be tested and the goal of the research is to investigate the method of bond gap control rather than the surface preparation, it was assumed that the recommended surface preparation will be used. It consisted of grinding the surface with sandpaper starting with a P240 gradation ending with P400 on a lathe for about 30 s each. This can be repeated on an industrial scale with a specially designed tool. Ultimately, the surfaces have been cleaned with acetone and dried with oil-free compressed air.

The curing process for both 3425 and 9466 has been realized with a one-step process of curing the epoxy at 80 °C for 2 h. Curing at elevated temperatures also protects from another important factor—hot strength—which influences the adhesive strength at a given temperature. It can be seen in technical datasheets [21,22] that the strength of adhesives cured at room temperature decreases by a half at 40 °C. Since the temperature expected during the operation of wishbones definitely exceeds this value, we decided to cure the epoxy at an elevated temperature, which should increase the glass transition temperature. One must keep in mind that the thermal treatments can influence the strength of the adhesive bond [30,31].

### 2.5. Test Setup

All tests were performed using an Instron 8512 testing machine (Instron Worldwide, Norwood, MA, USA). The testing speed was equal to 1 mm/s with a sampling rate of 1 KHz. This speed corresponds to a force increase of 10 kN per second, which should be similar to the rate of force increase in wishbones during vehicle operation. Obviously, there can be a difference between static and dynamic toughness as seen in [32], yet for the final project it is more important to be physically accurate with the testing method rather than obtain possibly higher results. The testing setup can be seen in Figure 3.

## 3. Results

The results were presented for each adhesive separately. Four types of samples were tested for each adhesive. The four samples type were:Inserts with 14 mm tubes utilizing geometrical solution of bond gap control;Inserts with 14 mm tubes utilizing glass beads solution of bond gap control;Inserts with 18 mm tubes utilizing geometrical solution of bond gap control;Inserts with 18 mm tubes utilizing glass beads solution of bond gap control;

The results of average force as a function of displacement for the two types of epoxy adhesive can be seen in Figure 4 and Figure 5. The results are presented for the samples closest to the average result in each group. This, however, does not depict the variation of the results, which will be presented later using whiskers boxplots in Figure 6 and Figure 7. They present the dispersion of ultimate tensile strength, and the average values are stored in Table 4. It is crucial to remember that the given values of “stress” are calculated as force per bond area, which is not the real stress which occurs in the bond. This is performed to compare the results to the values given in the technical datasheets and to give a general reference of results.

The failure mode identification of a designed bond is also a key element of the design validation. Figure 8 shows four samples chosen from samples of EA 3425, which failed during testing. The two on the right side represent tests performed on the beads solution of gap thickness control, and the two on the left represent the geometrical solution. What can be noticed is that in the first case (right), there is a layer of carbon fiber left on a sample. This would suggest that the samples failed in the adherend failure mode, meaning that the strength of the carbon fiber layers in the tube was insufficient to withstand the stress and broke off from the laminate. In second case (left) the failure mode was most likely interfacial or adhesive failure as the inserts were pulled out of the tubes, with adhesive stuck to the aluminum insert.

## 4. Discussion

The results obtained using the EA 9466 show that the adhesive joint is stronger in case of the beads gap control method. However, the spread of the results suggests that the performance of the bond is highly influenced by the quality of the manufacturing method. Of course, the samples were prepared with the same surface treatment and assembly methods, yet since the process is carried out by hand, it is obvious that some errors might occur.

The results obtained with EA 3425 indicate that the beads gap control method provides much better and repeatable results in comparison to the geometrical method. The spread of results is smaller relative to the geometrical method, and the obtained mean values are about twice as high. It would seem that the 3425 adhesive is better suited for the beads method of bond gap control. The authors’ explanation is that the viscosity of this adhesive, which causes it to be thixotropic, helps the epoxy to stay attached to the walls of the cylinder. This provides more adhesive in the bond and, in general, a better bond. The EA9466 has better results when the bond is properly filled with adhesive (even reaching 35 kN), but due to the low viscosity it is prone to escaping from the bond gap. Obviously, this is only a theory; there are more factors than can play a significant role such as the repeatability of the assembly method, etc.

Another important factor to consider was the failure mode of the joint. The preferred failure mode in engineering is the adherend failure mode [33] as it means that the bond is stronger than the material itself. Proper identification of the failure mode is crucial as it determines whether the design is valid [34]. As was seen in Figure 8, the interfacial layer of carbon in the tube was not strong enough and failed in case of the beads solution. This probably means that the interfacial stresses were very high locally, thus causing a failure. Future designs (to optimize the structure) should probably focus on developing a double lap joint and pay more attention to interfacial shear stress analysis as carried out in [35].

## 5. Conclusions

The research allowed us to determine the best adhesive and bond gap control method for a given design of an adhesive joint. This does not mean it will be optimal for all other adhesive joints, yet it gives some idea as to which system could be used for similar projects. The design loads of 8 kN and 12 kN for the 14 and 18 mm insert, respectively, are met with a safety coefficient for the manufacturing process of about 2. Obviously, the structure could be optimized further in terms of mass by decreasing the bond length. Another possibility is to use the double lap joint method by adding external bushing. This approach to adhesive joints results in no peeling stress as the joint is loaded symmetrically and no moment is created.

The geometrical method of bond gap control could be improved by inserting the adhesive externally under pressure by some opening in a tube or channels inside the insert. This would unfortunately also increase the cost of the manufacturing process. Additionally, the geometrical solution is harder to manufacture as some inserts in the final design require machining on five-axis milling machines, since the inserts are at some angles relative to reach other, which limits the possibility of using lathes.

The influence of testing the surface preparation was not investigated as it will be carried out using the adhesive manufacturers’ recommended process, which includes using sandpaper and thoroughly cleaning the surface. Obviously, the etching of the aluminum surface could be performed as it was reported to bring additional performance [12]. This can potentially increase the bonding strength to the aluminium surface, yet it is not where the debonding takes place as it happens at the interface between the adhesive and carbon fiber tube.

Comparing the results to [12], it can be seen that the beads method has slightly better results. Of course, the geometry was slightly different, yet if average “stress” is compared the results slightly better. On the other hand, the results obtained in [11] are higher by about 10% than the results obtained in this research. Again, it is difficult to directly compare as the geometries were slightly different and the used adhesives were not the same (although similar). The most important finding is that the method of glass beads can produce more stable results for a given bond, which allows for less safety coefficients as the worst-case scenarios are very close to average results.

To conclude, a few words about the final design of the wishbones should be said. Although they were not included in the Results section, they have been prepared using the 3425 and beads method of gap control. They have been assembled using a simple 10-mm-thick plate with machined holes for the positioning of shoulder screws. The whole assembly has been cured together, and then the wishbones were removed for final inspection. Since it is crucial that the wishbones do not fail during operation in the vehicle, some non-destructive testing would be helpful. To perform this, the wishbones were mounted in the machine used during the testing of the specimens and loaded to the maximum expected load value during operation. All the wishbones passed the proof test successfully. Other NDT methods are also available, such as those described in [36], yet this one has been chosen as the wishbones will experience very few cycles (one short race), but of high amplitude.

## Figures and Tables

**Figure 1 materials-14-01977-f001:**
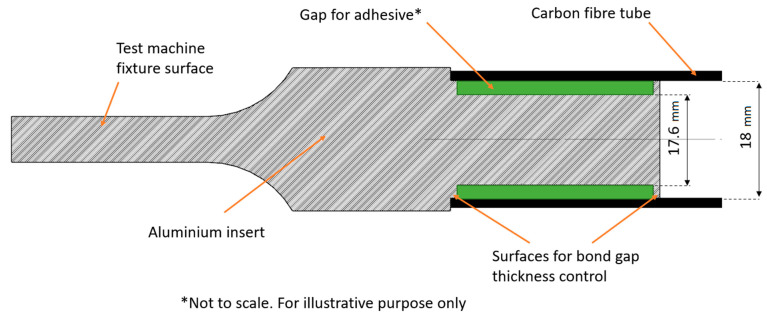
Idea of geometrical solution of bond gap thickness control.

**Figure 2 materials-14-01977-f002:**
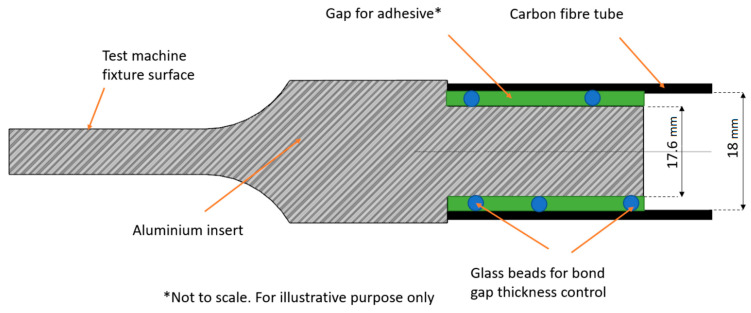
Idea of glass beads solution of bond gap thickness control.

**Figure 3 materials-14-01977-f003:**
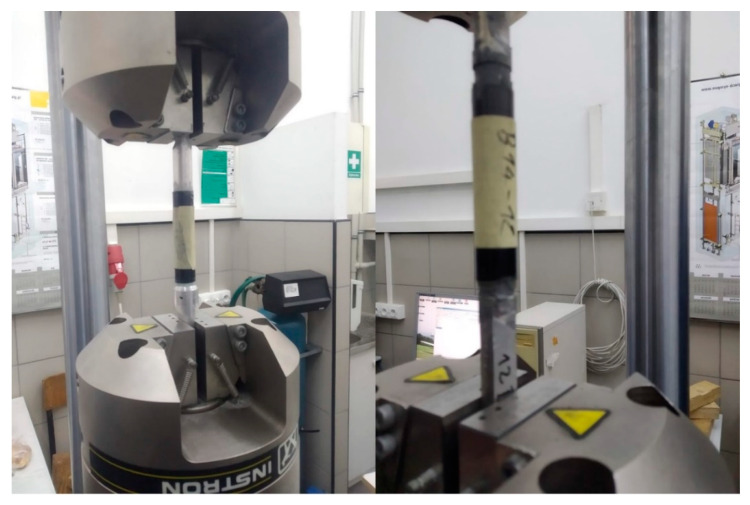
Test setup.

**Figure 4 materials-14-01977-f004:**
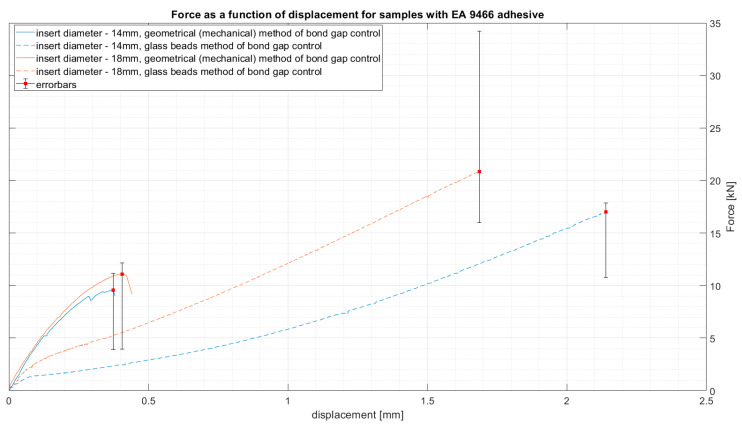
Force as a function of displacement for samples with EA 9466 adhesive.

**Figure 5 materials-14-01977-f005:**
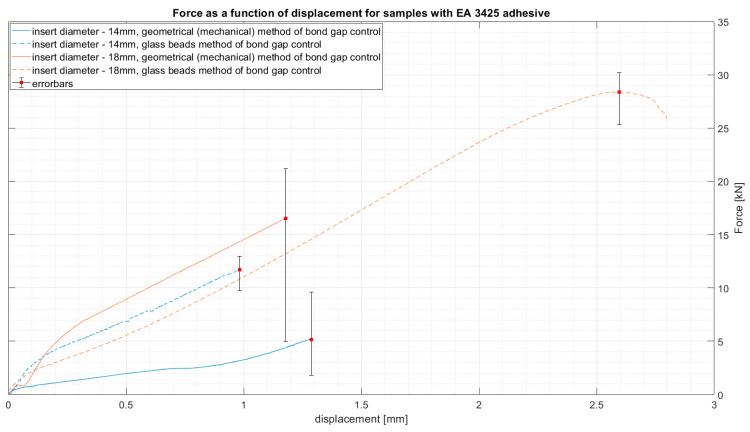
Force as a function of displacement for samples with EA 3425 adhesive.

**Figure 6 materials-14-01977-f006:**
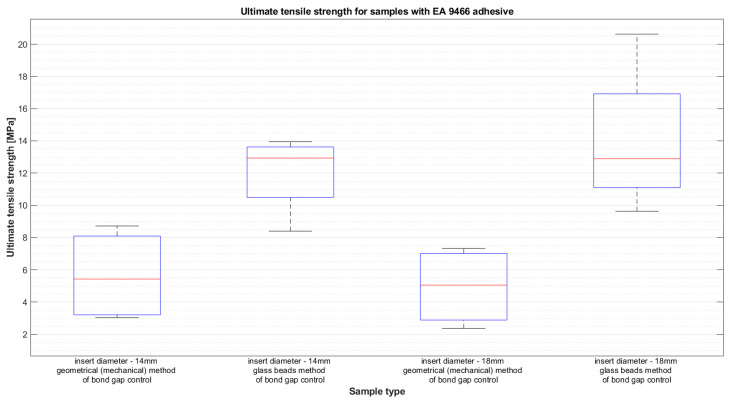
Ultimate tensile strength for samples with EA 9466 adhesive.

**Figure 7 materials-14-01977-f007:**
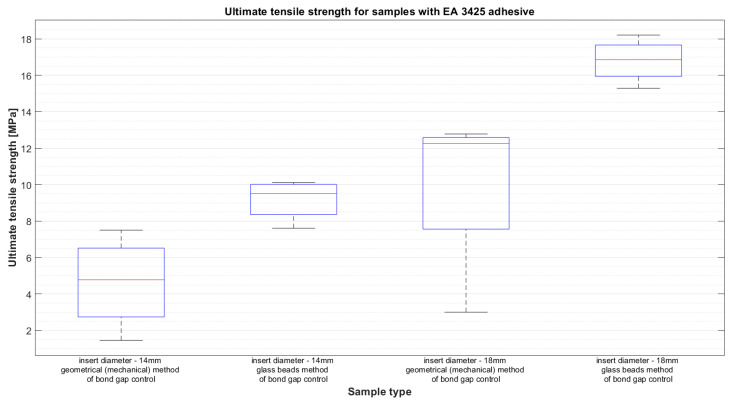
Ultimate tensile strength for samples with EA 3425 adhesive.

**Figure 8 materials-14-01977-f008:**
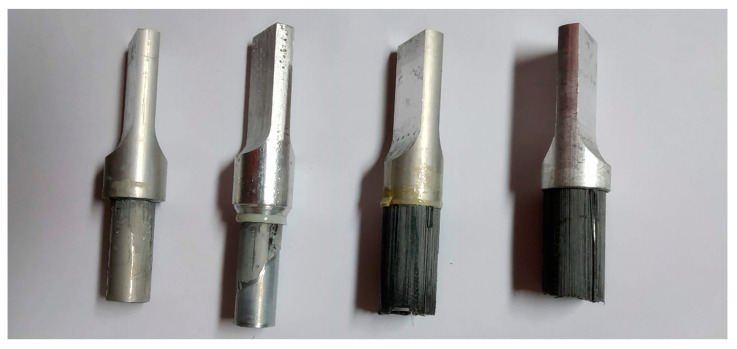
Failure modes of the adhesive bond—for geometrical solution on 3425 (two on the **left**) and beads solution (two on the **right**).

**Table 1 materials-14-01977-t001:** Parameters of carbon fiber-wrapped tubes [19].

**Outside OD (Inside ID) Diameter (mm)**	**16.7 (14)**	**20.8 (18)**
Tensile Strength (Axial) (MPa)	750	750
Youngs Modulus (Axial) (GPa)	90	90
Tensile Strength (Radially) (MPa)	600	600
Youngs Modulus (Radially) (GPa)	19	19
OD Tollerance (mm)	0.3	0.3
ID Tollerance (mm)	0.2	0.2
Major Poisson’s Ratio	0.14	0.14

**Table 2 materials-14-01977-t002:** Properties of adhesives used in the experiments taken from datasheets [21,22].

**Adhesive**	**EA 3425**	**EA 9466**
Tensile Modulus, ISO 527-3 (MPa)	1350	1718
Elongation ISO 527-3, %	3	3
Lap Shear Strength ISO 4587		
On Aluminum (abraded) (MPa)	7–13	26
On Glass reinforced Polymer (MPa)	0.6–1.2	5
On Glass Fiber Reinforced Epoxy (MPa)	-	-

**Table 3 materials-14-01977-t003:** Adhesion models comparison.

**Model**	**Advantage**	**Disadvantage**
Kendall	Simple, few data required to obtain result	Inaccurate, predicts only peel strength/stress
Golland-Reissner	Predicts peel and shear strength/stress, good for initial design purposes	Complicated, does not match some of the experimental results correctly
Experimental	Probably best accuracy	Expensive as samples of the exact geometry must be prepared and tested

**Table 4 materials-14-01977-t004:** Average results of ultimate “stress”.

**Adhesive**	**14 mm** **Geometrical**	**14 mm** **Beads**	**18 mm** **Geometrical**	**18 mm** **Beads**
Average stress EA 9466 (MPa)	5.65	12.05	4.95	14.01
Average stress EA 3425 (MPa)	4.62	9.18	10.07	16.80

## Data Availability

The data presented in this study are available on request from the corresponding author.

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
