# Peer review of "Evaluation of Bonding Gap Control Methods for an Epoxy Adhesive Joint of Carbon Fiber Tubes and Aluminum Alloy Inserts"

_materials, 2021, doi:10.3390/ma14081977_

Round 1
Reviewer 1 Report
There are some weaknesses through the manuscript which need improvement. Therefore, the submitted manuscript cannot be accepted for publication in this form, but it has a chance of acceptance after a major revision. My comments and suggestions are as follows:
1- Abstract gives information on the main feature of the performed study, but some details about the bonded joints must be added.
2- Authors must clarify necessity of the performed research. Aims and objectives of the study, and also differences with the previous researches must be clearly mentioned in the last part of introduction.
3- The literature study must be enriched. In this respect, authors must read and refer to the following papers: (a) https://doi.org/10.4028/www.scientific.net/KEM.789.45 (b) https://doi.org/10.1016/j.compositesb.2018.08.006 Also, more experimental figures are welcome.
4- References for the values mentioned in Table 1 and 2 are required.
5- It would be nice, if authors add a table and briefly describe benefit, disadvantage, and limitations of different adhesive bond models in 2.2.
6- It is recommended to prepare curves (e.g., Fig. 3-6) in AMTLAB environment or present in s more scientific way.
7- Illustrating stress-strain curves of the examined specimens is a necessity.
8- In its language layer, the manuscript should be considered for English language editing. There are sentences which have to be rewritten.
9- The conclusion must be more than just a summary of the manuscript. List of references must be updated based on the proposed papers. Please provide all changes by red color in the revised version.
Author Response
There are some weaknesses through the manuscript which need improvement. Therefore, the submitted manuscript cannot be accepted for publication in this form, but it has a chance of acceptance after a major revision. My comments and suggestions are as follows:
1- Abstract gives information on the main feature of the performed study, but some details about the bonded joints must be added.
The abstract has been changed accordingly. It contains information about adhesive type, bond gap, length and diameter of an insert/tube
2- Authors must clarify necessity of the performed research. Aims and objectives of the study, and also differences with the previous researches must be clearly mentioned in the last part of introduction.
The information about differences with previous research has been added. It is mainly the method of controlling the bond gap. The introduction part has been rewritten.
3- The literature study must be enriched. In this respect, authors must read and refer to the following papers: (a) https://doi.org/10.4028/www.scientific.net/KEM.789.45 (b) https://doi.org/10.1016/j.compositesb.2018.08.006 Also, more experimental figures are welcome.
Thank you for recommendations. The literature has been enlarged by the mentioned titles and a few others. Figures 1 and 2 have been redrawn to show the concept a bit better.
4- References for the values mentioned in Table 1 and 2 are required.
Done. Line 126
5- It would be nice, if authors add a table and briefly describe benefit, disadvantage, and limitations of different adhesive bond models in 2.2.
Table 3 has been added to the main text. Line 204
6- It is recommended to prepare curves (e.g., Fig. 3-6) in AMTLAB environment or present in s more scientific way.
Force/displacement graphs have been added. This is popular method of presenting results among other papers. The stress strain representation might not be correct as the stress is non-uniformly distributed (highest around the ends). We have tried to measure strain with digital image correlation methods, but since the bond is on the inside of a tube it is quite tricky. Measuring relative position of insert and tube and then measuring the difference does not produce sensible results. Thus, the displacement has been taken from the measuring apparatus (machine) itself. We tried to calculate the strain from that but since there are some compliant parts and it is not normalized test the results are not accurate (e.g. displacement of 3mm would mean very high value of strain since the bond has 30mm, but obviously it is not what happens in the bond).
The figures with whiskers boxplot present statistical distribution of obtained results. They are quite often used in the reports. We have decided that it will be better to show the spread and range of data. It is more useful than presenting mean value since the mean value doesn’t say anything about population.
7- Illustrating stress-strain curves of the examined specimens is a necessity.
As explained above
8- In its language layer, the manuscript should be considered for English language editing. There are sentences which have to be rewritten.
Understood
9- The conclusion must be more than just a summary of the manuscript. List of references must be updated based on the proposed papers. Please provide all changes by red color in the revised version.
We have used the “track changes” option in MS word as we were asked to do this by editorial. But I believe that shows exactly what kind of modifications were made.
Reviewer 2 Report
The introduction is very briefly and not discussed actually which re the main challenges ;
Also the authors were focused very narrow and not looked over other innovative solutions; just as suggestion please take a look on https://doi.org/10.1016/j.ijadhadh.2020.102718
The novelty should be better emphasized otherwise it will be difficult for readers to understand what is novel in this study. Also not sure if a smaller bonding thickness is the best as there was no any critical review in this respect
“curing at elevated temperature” which are these temperatures ??for how long ?
“temperature increases their final strength properties” ok please provide details how much and values
All acronyms should eb described before their first appearance e.g. HRT
“grinding the surface with sandpaper” Ok, but how easy can be performed at industrial scale and with reproducibility ???
Details of each part in the setup from Figure 3
Please check you have twice noted Figure 3 and 4
You present the Figure 3-8 but they were not introduced in text and not discussed at all
Please provide details for each “4 random samples that failed during testing”
What does mean 14geo? And other notations ?
I think this “adherend failure mode” is just a speculation as long as you don’t show any evidence
The discussion and the results part should be extended and carefully presented against literature data
I have noted only very few references from recent years, please update this with more recent one
Author Response
The introduction is very briefly and not discussed actually which re the main challenges ;
Also the authors were focused very narrow and not looked over other innovative solutions; just as suggestion please take a look on https://doi.org/10.1016/j.ijadhadh.2020.102718
Thank you for recommendation. The suggested title has been added along with a few other papers.
The novelty should be better emphasized otherwise it will be difficult for readers to understand what is novel in this study. Also not sure if a smaller bonding thickness is the best as there was no any critical review in this respect
Two paragraphs were added to the introduction, we hope that they will bring brighten the picture of the scope and purpose of the research.
Smaller bonding thickness is contradictory with the adhesive models such as Golland Reissner model and in fact it is a subject of a debate currently. Indeed, more studies and research in this area is needed. The studies [11] and [12] which are closest to the type of bond we are investigating were using this 0.2mm bond thickness therefore we have decided to use it and as a novelty introduce other method of bond gap control.
“curing at elevated temperature” which are these temperatures ??for how long ?
EA 3425, EA 9466 cured at 80ËšC for 2h and EA 9514 cured at 120ËšC for 2h. The abstract has been changed accordingly to show those parameters as well as main text
“temperature increases their final strength properties” ok please provide details how much and values
Done in lines from 130-132. It is about 10% if cured at 80C. This information has been added to the text.
All acronyms should eb described before their first appearance e.g. HRT
Done, some like EA were not changed at it is just a trading name of a product
“grinding the surface with sandpaper” Ok, but how easy can be performed at industrial scale and with reproducibility ???
It has been done on the lathe. This was not mentioned in the text, but now it is. As far as the industrial application goes probably some kind of a tool would have to be designed. (Lines
Details of each part in the setup from Figure 3
Both figures 3 and 4 have been redrawn
Please check you have twice noted Figure 3 and 4
Yes, my mistake. It has been fixed.
You present the Figure 3-8 but they were not introduced in text and not discussed at all
The results part has been re-written. All notations and figures are now introduced.
Please provide details for each “4 random samples that failed during testing”
It was bad translation from my side. What I wanted to say is that the type of depicted samples was chosen randomly to show what kind of failure was observed for most of them. The samples were average samples from that group.
What does mean 14geo? And other notations ?
The results section has been re-written and now explains the notations better
I think this “adherend failure mode” is just a speculation as long as you don’t show any evidence
I think so too. We didn’t perform any microscope tests it is rather interesting fact that can be investigated in further study. Maybe we will delete this section.
The discussion and the results part should be extended and carefully presented against literature data
We are benchmarking mainly against [11] and [12] as they have performed projects closest to the one that we are preparing. We have mostly focused at comparing the research with their results. Perhaps we could compare it somehow to other results, but we would need to figure out how to do this as they are a bit different.
I have noted only very few references from recent years, please update this with more recent one
Some suggested references were added.
Round 2
Reviewer 1 Report
The paper has been improved and corresponding modifications have been conducted. In my opinion, the current version can be considered for publication.
Author Response
We would like to thank the reviewer for his/her valuable comments.
Reviewer 2 Report
.
Author Response

(The authors gave the same response as above.)
